# The Clock:Cycle complex is a major transcriptional regulator of *Drosophila* photoreceptors that protects the eye from retinal degeneration and oxidative stress

Juan Jauregui-Lozano[1], Hana Hall[1,2], Sarah C. Stanhope[1], Kimaya Bakhle[1], Makayla M. Marlin[1], Vikki M. Weake[1,2,3]*

1 Department of Biochemistry, Purdue University, West Lafayette, Indiana, United States of America,
2 Purdue Institute for Integrative Neuroscience, Purdue University, West Lafayette, Indiana, United States of America, 3 Purdue Center for Cancer Research, Purdue University, West Lafayette, Indiana, United States of America

* vweake@purdue.edu

**Data Availability Statement:** All data are fully available without restriction. All the high-throughput data is available at the Gene Expression

## Abstract

The aging eye experiences physiological changes that include decreased visual function and increased risk of retinal degeneration. Although there are transcriptomic signatures in the aging retina that correlate with these physiological changes, the gene regulatory mechanisms that contribute to cellular homeostasis during aging remain to be determined. Here, we integrated ATAC-seq and RNA-seq data to identify 57 transcription factors that showed differential activity in aging *Drosophila* photoreceptors. These 57 age-regulated transcription factors include two circadian regulators, Clock and Cycle, that showed sustained increased activity during aging. When we disrupted the Clock:Cycle complex by expressing a dominant negative version of Clock (Clk$^{DN}$) in adult photoreceptors, we observed changes in expression of 15–20% of genes including key components of the phototransduction machinery and many eye-specific transcription factors. Using ATAC-seq, we showed that expression of Clk$^{DN}$ in photoreceptors leads to changes in activity of 37 transcription factors and causes a progressive decrease in global levels of chromatin accessibility in photoreceptors. Supporting a key role for Clock-dependent transcription in the eye, expression of Clk$^{DN}$ in photoreceptors also induced light-dependent retinal degeneration and increased oxidative stress, independent of light exposure. Together, our data suggests that the circadian regulators Clock and Cycle act as neuroprotective factors in the aging eye by directing gene regulatory networks that maintain expression of the phototransduction machinery and counteract oxidative stress.

## Author summary

Age-associated changes to the retinal transcriptome often correlate with physiological changes, such as loss of visual function and increase in cell death. However, the

Omnibus (GEO) repository under the specific accession codes outlined below: GSE169328 (D10 ATAC-seq and RNA-seq), GSE184069 (D20 – D60 ATAC-seq), GSE174515 (D20 – D60 and D10 light/dark RNA-seq), GSE183746 (ClkDN ATAC-seq), GSE184121 (ClkDN RNA-seq). Numerical data underlying graphs are provided in S4 Table and S5 Table.

**Funding:** This research was funded by the National Eye Institute of the NIH under Award Number R01EY024905 to V.M.W and by the Bird Stair Research Fellowship and Ross Lynn Research Scholar fund (Biochemistry Department, Purdue University) to J. J-L. and by NIH training award 5T32GM125620 to S.C.S. The funders had no role in study design, data collection and analysis, decision to publish, or preparation of the manuscript.

**Competing interests:** The authors have declared that no competing interests exist.

mechanisms that contribute to these transcriptomic changes are poorly understood. Here, we used a genomics/bioinformatics approach to identify transcription factor binding sites with differential activity in aging *Drosophila* retina outer photoreceptors. Amongst these age-regulated transcription factors, we identify the circadian regulators Clock and Cycle. Using a genetics approach, we find that photoreceptor-specific disruption of the Clock:Cycle complex makes the *Drosophila* eye susceptible to light-dependent retinal degeneration, and light-independent increase of oxidative stress, showing that a functional circadian clock contributes to visual health and function in *Drosophila*. Because disruption of circadian rhythms has been associated with the onset of several age-related eye diseases, our data shows that the *Drosophila* retina serves as a useful model to study how disruption of the circadian clock contributes to neurodegeneration in the retina.

## Introduction

One of the hallmarks of the aging eye, as well as many age-related eye diseases, is the loss of photoreceptor function and survival [1,2]. The aging epigenome and transcriptome of cells in the retina undergo changes that correlate with decreased visual function and increased cell death [3–5]. Importantly, disruption of epigenetic mechanisms is associated with the onset of age-related eye diseases, such as age-related macular degeneration [6,7], suggesting that transcriptional regulation contributes to the changes in homeostasis that are observed in the aging eye. However, we still have only a basic understanding of how the molecular mechanisms that drive the age-associated changes in the transcriptome increase the risk of ocular disease with advanced age.

Transcription factors (TF) function as regulatory hubs of gene expression programs in a highly-tissue specific manner. While several conserved pathways contribute to changes in the aging transcriptome across tissues, such as the longevity-associated FOXO and Insulin axis [8–10], age-associated changes in gene regulatory networks can be highly specific to individual cell types [5,11]. The emergence of bioinformatic and computational approaches that combine chromatin accessibility data with transcription factor binding sites (TFBS) has allowed researchers to interrogate unbiasedly how transcription factor activity changes at a genome-wide scale in different biological conditions by estimating changes in chromatin accessibility around TFBS (see [12] for review on chromatin accessibility data analysis). The majority of chromatin accessibility studies in the eye have focused on development in invertebrate [13] or vertebrate models [14], and TF activity in the developing eye has often been assessed based on transcriptomic-based co-expression inference rather than chromatin accessibility [15]. Further, ATAC-seq analysis of retinal pigmented epithelium from patients with age-related macular degeneration identified global changes in chromatin accessibility at the onset of the disease state, suggesting that differential activity of regulatory elements strongly contributes to the initial stages of age-associated ocular disease [16]. Thus, identification of TFs with differential activity in the aging eye could provide insight into the mechanisms that contribute to the increased risk of retinal degeneration during aging.

*Drosophila melanogaster*, like humans, experience an age-dependent decline in visual function coupled with increased risk of retinal degeneration [3,17]. Although flies possess a compound eye that differs anatomically from the vertebrate eye, much of the phototransduction machinery is conserved between flies and mammals, with the outer fly photoreceptors resembling vertebrate rods in function [18,19]. There are six outer photoreceptors (R1 –R6 cells)

within each ommatidium that express the light-sensing protein Rhodopsin 1 (Rh1) and are responsible for motion vision and phototaxis, which decline with age [3,17]. Considering its relatively short lifespan, *Drosophila* provides a useful model system for studying the processes involved in normal aging within specific tissues, such as the eye [20]. To profile the transcriptome and epigenome of specific cell types in the eye, we have developed a cell type-specific nuclei immunoenrichment technique that we have previously used to examine gene expression in aging photoreceptors [3,21,22]. Here, we applied this technique to profile the transcriptome and chromatin accessibility landscapes of *Drosophila* photoreceptors across an extended time course into relatively old age. By integrating these aging data from photoreceptors, we identified 57 TF motifs that were differentially accessible during aging, suggesting age-dependent changes in TF activity. Amongst these "age-regulated TFs", we identified the binding motif of the master circadian regulators, Clock (Clk) and Cycle (Cyc), which showed sustained increases in activity with age. Using a dominant negative mutant of Clock (Clk[DN]) that disrupts the Clk:Cyc complex and abolishes rhythmic transcription, we showed that the Clk:Cyc complex has an integral role in controlling gene expression of 15–20% of active genes, and maintaining global levels of chromatin accessibility in photoreceptors. Further, we show that expression of Clk[DN] in photoreceptors leads to progressive retinal degeneration, which was suppressed when flies were reared in constant darkness. Our data identify a novel neuroprotective role for the circadian clock in the *Drosophila* eye, and suggest that this role may become increasingly critical in advanced age to prevent retinal degeneration.

## Results

### Tissue-specific profiling of the photoreceptor nuclear transcriptome reveals significant changes during early and late aging

To profile the transcriptome of aging *Drosophila* photoreceptors, we used our recently improved tissue-specific nuclei immunoenrichment approach [22]. Briefly, we tag nuclei using a Green Fluorescent Protein (GFP) fused to the "Klarsicht, ANC-1, Syne Homology" (KASH) domain, which anchors GFP to the outer nuclear membrane and allows for nuclei purification; GFP[KASH] is expressed in outer photoreceptors under Rh1-Gal4 control, herein referred as Rh1>GFP[KASH] (Fig 1A). Similar to wild-type strains, such as OregonR and Canton-S [23], Rh1>GFP[KASH] flies begin to show a substantial decline in viability after day 60, with 50% of male flies dying by day 70 [3]. Thus, to generate a comprehensive aging dataset, we purified photoreceptor nuclei over 10-day windows from day 10 until day 60 and performed RNA-seq using 400 age-matched male flies per biological replicate (Fig 1B).

We assessed the overall variability of the RNA-seq samples using Principal component analysis (PCA). The RNA-seq samples clustered together with 48.3% of the variation separating the samples by age in a progressive manner (Fig 1C). We also observed a similar grouping by age using Pearson's correlation analysis with high concordance between the three biological replicates (Pearson's $r > 0.91$) (S1 Fig). These observations reveal progressive changes in transcription during aging in photoreceptors, similar to previous studies from our group that aged flies to day 40 (D40) [3]. Differential expression analysis of each time point relative to day 10 (D10), the youngest state, revealed significant changes in gene expression during aging (p-adj<0.05) (Fig 1D and S1 Table). Hierarchical clustering of the genes that were differentially expressed between any age and D10 revealed that 1412 and 982 genes showed progressive increases, or decreases in expression, respectively. Gene Ontology (GO) term analysis revealed that the genes that were age-upregulated (Clusters 2 and 3) were enriched for several processes including cytoplasmic translation and double strand break repair (Fig 1E). In contrast, the genes that were age-downregulated (Cluster 5) were enriched for neuronal processes, such as

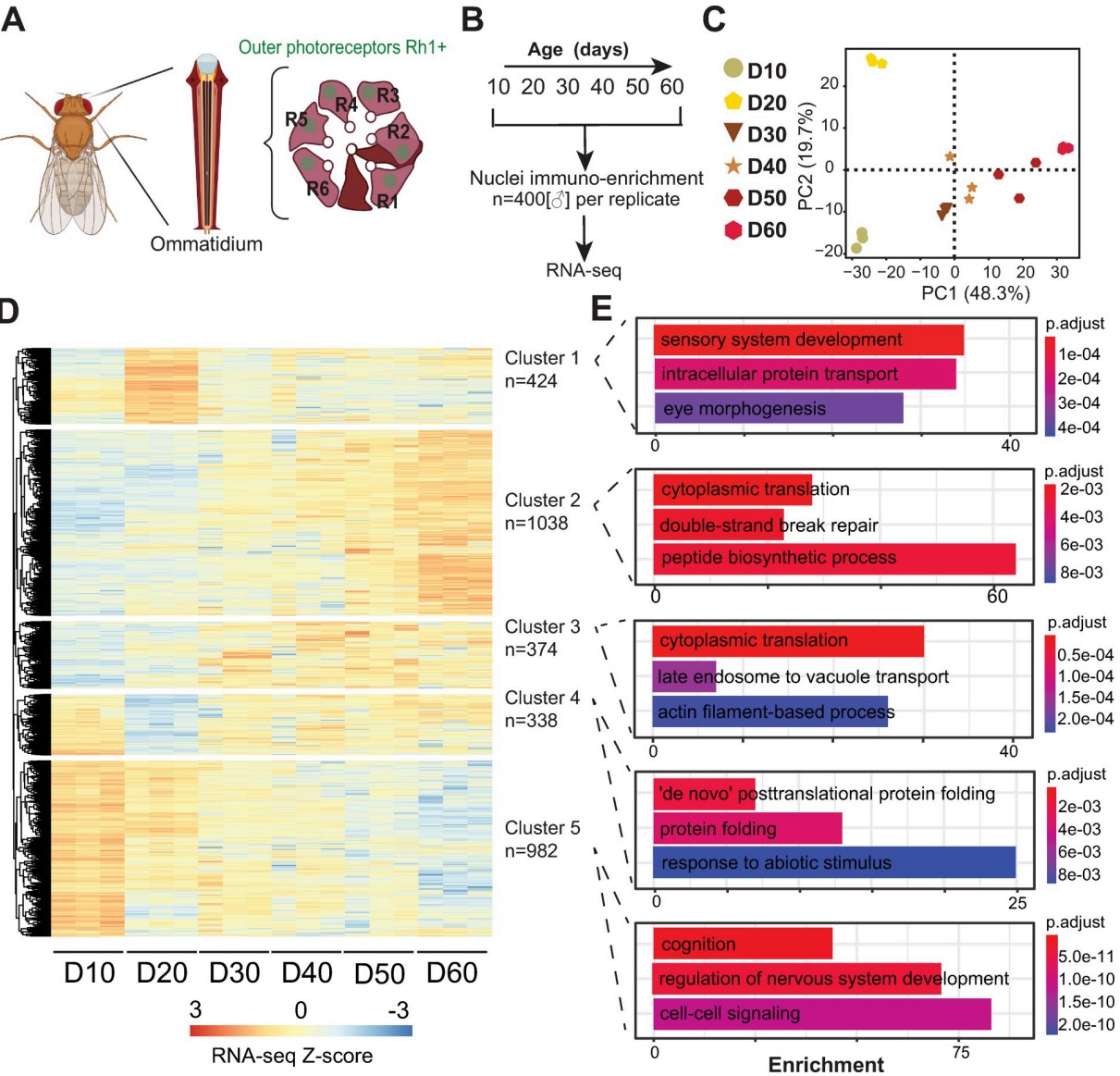

**Fig 1. Tissue-specific profiling of the photoreceptor nuclear transcriptome reveals significant changes in early and late aging.** A. Schematic of the *Drosophila* compound eye, composed of ommatidia that contain six outer photoreceptors that express Rhodopsin 1 (Rh1). Outer photoreceptor nuclei are labeled with GFP$^{KASH}$. B. Experimental design for the aging RNA-seq time course. C. Principal Component Analysis (PCA) of aging RNA-seq samples based on gene counts. D. Hierarchically clustered heatmap of aging RNA-seq samples. Only genes that were identified as being differentially expressed in any condition relative to D10 are shown. Z-scores are calculated based on normalized counts obtained using DESeq2, and the heatmap is divided into five clusters based on the dendrogram. E. Gene Ontology (GO) term analysis of significantly enriched genes in each cluster.

cognition and regulation of nervous system development. Although 75% of the differentially expressed genes (2394 out of 3156) exhibit progressive changes, we also identified two clusters with expression changes only during early aging (D20; Clusters 1 and 4). We note that while D20 was separated by age from the other samples along PC1, it was also separated from its nearest time points (D10 and D30) along PC2 (Fig 1C), consistent with these early and transient changes in gene expression in aging photoreceptors. Interestingly, Cluster 4, which contained genes that showed an early decrease in transcription but increased levels again at later time points, was enriched for protein folding. These age-associated transcriptomic signatures

are consistent with our previous observations in aging photoreceptors and with aging studies from other tissues [3,4].

## Photoreceptors undergo substantial changes in transcription factor activity during aging

To identify mechanisms that contribute to the age-associated transcriptomic changes in photoreceptors, we sought to evaluate changes in TF activity. TFs regulate transcriptional outputs of their targets by acting as activators or repressors of gene expression at the transcriptional level. Importantly, TF activity can be affected by several factors, including protein levels, localization, and post-translational modifications [24,25]. Because of the technical complexity of isolating intact photoreceptors for proteomic studies, and the relatively low protein abundance of TFs in the eye [26], we used our RNA-seq data to assess how transcript levels of genes associated with the GO category "DNA-binding transcription factor activity, RNA polymerase II-specific" (GO:0000981) changed during aging. Notably, 23% of TFs showed significant differential expression during aging at the nuclear transcript level (Fig 2A), suggesting that differential TF activity could contribute to the aging transcriptome of photoreceptors. To identify TFs with differential activity during aging, we used *diffTF*, a software package that estimates genome-wide changes in TF motif/binding activity based on differences in aggregate ATAC-seq signal around predicted/validated TF binding sites, or TFBS [27]. We refer to TF motif/binding activity as "TF activity", as defined in the *diffTF* study [27]. Although *diffTF* analysis provides an estimate of TF activity, rather than a direct measurement of TF binding to DNA, *diffTF* was highly ranked as an approach to assay genome-wide changes in TF activity [28]. Thus, we first profiled the chromatin accessibility landscape of aging photoreceptors at the same time-points as for RNA-seq. PCA of ATAC-seq samples revealed that 69.7% of the variation could be explained by age (Fig 2B), and Pearson correlation analysis showed high concordance between biological replicates. Additionally, accessible peak annotation revealed a stable distribution of peaks through-out aging (S2A Fig). Together with the RNA-seq aging time course, these data indicate that the chromatin accessibility and transcriptional landscape of photoreceptors undergo progressive changes during aging. We then used *diffTF* to compare all time points relative to D10. The TFBS used for this analysis were generated by CIS-BP, which provides a comprehensive dataset of experimentally validated TFBS [29]. We also examined *de novo* and known motifs identified using Homer [30] (Fig 2C). We took advantage of the aging RNA-seq time series data to perform *diffTF* analysis in integrative mode, enabling us to discard TFs that were not expressed in adult photoreceptors. Using this approach, we identified 57 TFs whose binding sites showed significant differential activity during aging (FDR<0.001) (Fig 2C), herein referred to as "age-regulated TFBSs". Most of these age-regulated TFs showed continuous changes in activity with age, with nearly two thirds showing increased activity with age. We observed an almost two-fold increase in the number of age-regulated TFs identified at D50 and D60 relative to younger ages, suggesting that late aging is associated with distinct changes in gene regulatory networks (Fig 2D). Hierarchical clustering of these age-regulated TFs by the mean weighted difference between each age comparison resulted in grouping of TFs that are known to associate *in vivo*, such as Motif 1 Binding Protein (M1BP), DNA replication element factor (Dref) and Boundary element-associated factor of 32kD (BEAF32), which bind topologically associating domains [31]. We refer to TF proteins here using their gene name and symbol (non-italicized), and provide a complete list of all genes/proteins referred to in this study with their corresponding Flybase ID numbers in S3 Table. We also identified Moira (mor), which physically interacts with Similar (sima) [32]. Additionally, we identified Seven up (svp) and PvuII-PstI homology 13 (Pph13), which physically interact to regulate eye-specific

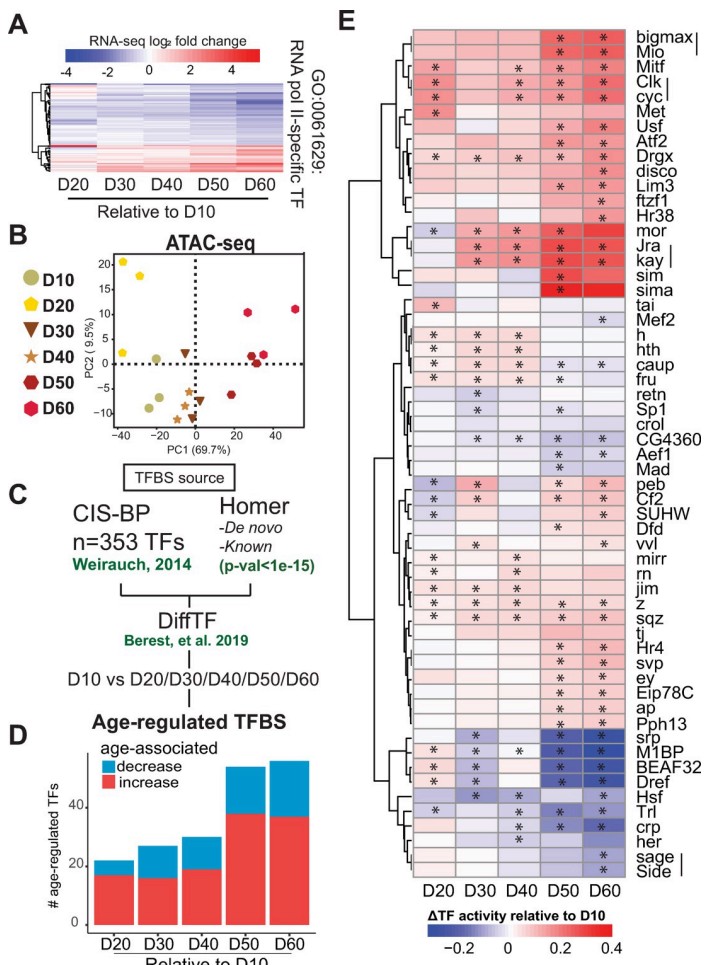

**Fig 2. Photoreceptors undergo substantial changes in TF activity during aging.** A. Heatmap of RNA-seq fold change during aging for significantly differentially expressed genes (DEGs; p-adj<0.05, |FC|>1.5 relative to D10) associated with the GO term "RNA polymerase II-specific DNA-binding transcription factor binding" (GO:0061629). Genes are hierarchically clustered by $\log_2$ fold change values. B. PCA of aging ATAC-seq samples based on the read distribution over binned genome. C. Schematic of differential transcription factor (TF) analysis approach using *diffTF*. D. Bar plot indicating the number of significant age-regulated TFs at each age relative to D10 (p-adj<0.001). E. Hierarchical clustered heatmap of age-regulated TFs with significant changes in activity between any age and D10 (asterisk, p-adj<0.01). Scale represents the relative change in TF activity with red showing higher TF activity in old samples relative to D10, and blue indicating an age-associated decrease in activity. TFs that bind a common motif and cluster together are indicated by a vertical line.

transcriptional programs together with Eyeless (ey) [33]. Considering that svp, ey, and Pph13 show modest but significant increases in TF activity with age, our data also indicate that photoreceptor identity remains distinct even at advanced age in flies.

The stress response is transcriptionally regulated during aging across a broad range of tissues [34]. Accordingly, we identified several TFs that are involved in the regulation of stress response genes as having differential activity with age including the *Drosophila* JUN-FOS complex that is formed by Jun-related antigen (Jra) and Kayak (kay) [35], the *Drosophila* homolog of Hypoxia response factor HIF-1a, sima [36], Heat shock factor (Hsf) [37], and Activating transcription factor-2 (Atf-2) [38,39]. We also identified Cropped (crp), which has previously been associated with aging through *in silico* analysis due to interactions with DNA repair pathways [40].

One of the most interesting changes in TF activity during aging was in the activity of the master circadian regulators Clock (Clk) and Cycle (Cyc), which showed a progressive increase in TF activity during aging. Clk and Cyc bind the same CACGTG motif upon heterodimerization, explaining the identification and clustering of both factors in the *diffTF* analysis, which uses DNA sequence motifs as the TFBS source (S2B Fig). The circadian clock is a molecular time keeper that controls rhythmic behaviors, such as locomotor behavior, which is coordinated by pacemaker neurons located in the brain [41]. In *Drosophila*, many peripheral tissues also contain working clocks, including the fat body, Malpighian tubule cells, and the retina [42]. Importantly, aging has been associated with changes in the circadian clock [43,44]. We note that flies for the aging time course experiment were raised under a 12:12 hour light-dark cycle, and were harvested at relative Zeitgeber time (ZT) 6 +/- 1 hour (see Materials and Methods), suggesting that the enrichment of Clk and Cyc in the *diffTF* analysis is not simply due to a time-of-day bias in sample collection. Taken together, our data identify a subset of TFs that exhibit significant changes in activity during aging in photoreceptors, including TFs associated with stress response and circadian rhythm.

## Clock regulates the transcriptional output of phototransduction genes in photoreceptors

Clk and Cyc are basic Helix-Loop-Helix (bHLH)-TFs that form a heterodimer (Clk:Cyc) to control rhythmic expression of their targets. Canonical transcriptional targets of the Clk:Cyc complex include core clock genes, such as *vrille* (*vri*), *PAR-domain protein 1* (*Pdp1*), *timeless* (*tim*), *period* (*per*), and *clockwork orange* (*cwo*). However, Clk and Cyc also regulate transcription of many other genes either directly or indirectly, and these genes are often referred as "Clock-output genes", and can include tissue-specific genes [45,46]. To evaluate the biological role of Clock-dependent transcription specifically in photoreceptors, we disrupted the Clk:Cyc complex by expressing a dominant negative version of Clk (Clk$^{DN}$) specifically in outer photoreceptors under Rh1-Gal4 control. Clk$^{DN}$ lacks the basic DNA binding domain, impairing its recruitment to DNA without disrupting its ability to form a heterodimer with Cyc (Fig 3A) [47]. Expression of Clk$^{DN}$ inhibits Clock-dependent transcription and rhythmic behaviors *in vivo* when expressed in pacemaker or antennal neurons [47]. To facilitate nuclei immuno-enrichment for RNA-seq analysis, we co-expressed UAS-GFP$^{KASH}$. As a control, we profiled the transcriptome of photoreceptors that expressed LacZ, herein referred as Rh1>Ctrl. We collected both Rh1>Clk$^{DN}$ and Rh1>Ctrl flies at ZT 9, when Clock-dependent transcription is active [48,49], harvesting flies at D1 and D10 to study the progressive effect of disrupting the circadian clock in adult photoreceptors (Fig 3B). We note that Rh1-Gal4 activity begins in the very late stages of pupal development [50]; thus Rh1>Clk$^{DN}$ flies have a disrupted Clk:Cyc complex in adult photoreceptors, but not in the developing eye.

Differential gene expression (DGE) analysis of Rh1>Clk$^{DN}$ relative to Rh1>Ctrl at either D1 or D10 revealed a consistent downregulation of direct Clk:Cyc targets such as *vri*, *per*, *tim* and *Pdp1* (Figs 3C and S2A and S2 Table). We also observed a significant upregulation of *Clk* itself, which we showed by qPCR reflects expression of *Clk$^{DN}$* rather than the endogenous wild-type *Clk* (S2B Fig). When we compared the control D1 and D10 flies, we did not observe differential expression of core clock genes (i.e. *vri*, *tim*, *per*), suggesting there is little change in the circadian clock in the early stages of adult life. In addition, only 147 genes changed in control flies between D1 and D10, indicating there is relatively little change in gene expression in general at these early stages of adult life in photoreceptors. In contrast, expression of Clk$^{DN}$ led to significant changes in expression of 15% and 22% of actively transcribed genes in photoreceptor at D1 and D10, respectively. These data demonstrate that continued expression of

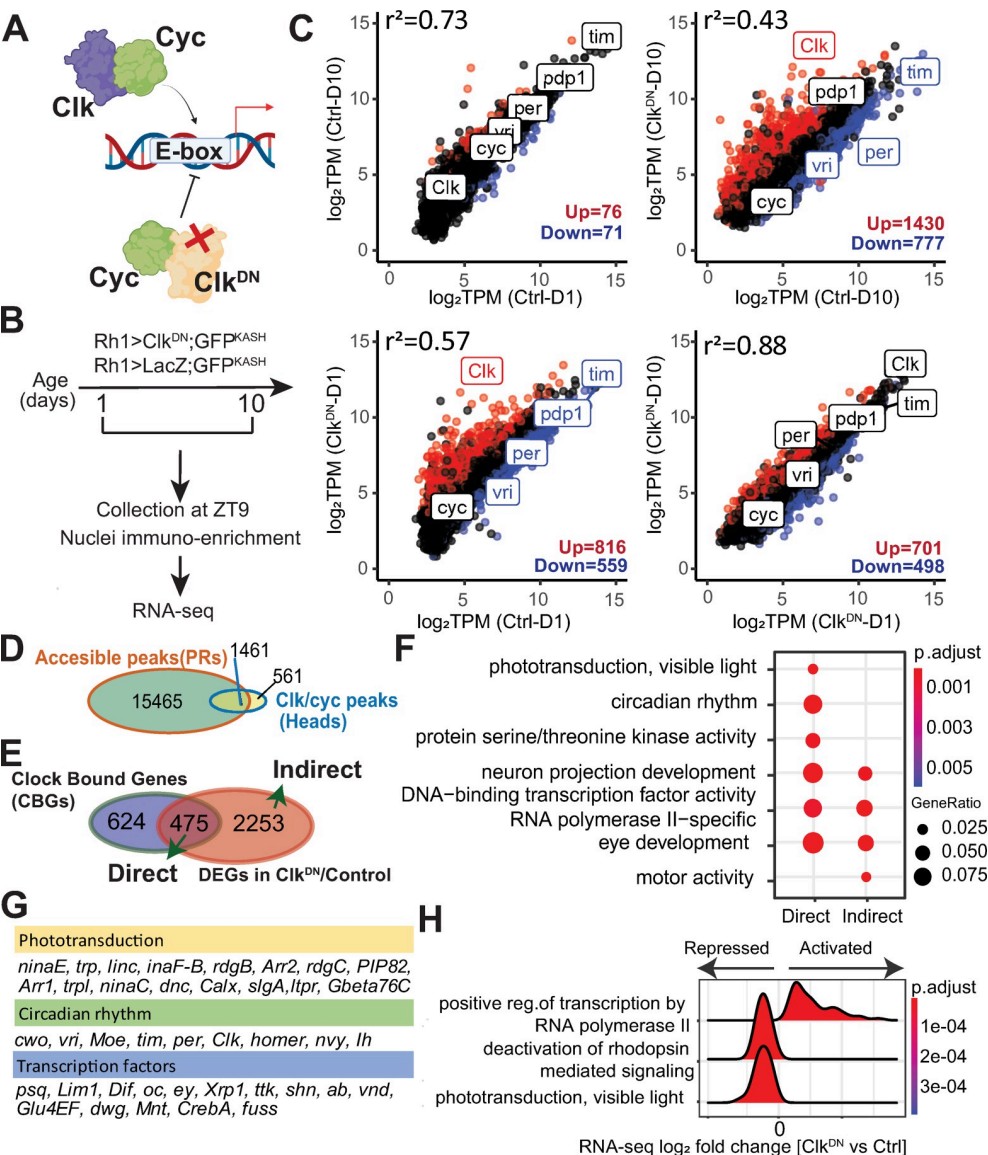

**Fig 3. Expression of Clk^DN in photoreceptors leads to genome-wide changes in TF activity and a widespread decrease in chromatin accessibility.** A. Schematic of Clk:Cyc activity. Expression of Clk^DN disrupts Clock-dependent transcription because it binds Cyc but does not contain the DNA binding domain. B. Schematic for the RNA-seq experiments. Flies that express the GFP^KASH tag and either Clk^DN or LacZ (control) in photoreceptors were aged to D1 or D10 prior to NIE. Flies were harvested at the indicated ZT. C. Scatter plot of mean expression (TPM, n = 3) for the indicated pair-wise comparisons. Up- and down-regulated DEGs (p-adj<0.05, |FC|>1.5) are colored in red and blue, respectively. Core clock genes *Clk*, *cyc*, *vri*, *per*, *Pdp1* and *tim* are highlighted. D. Genomic overlap of accessible peaks in photoreceptors and Clk:Cyc binding sites identified using ChIP-seq in [51]. E. Venn diagram comparing overlap of Clock-bound genes (CBGs-green) with genes that were differentially expressed (adj p <0.05, |FC|>1.5) in Rh1>Clk^DN relative to control. F. GO term analysis of Clock direct and indirect targets. G. Table showing selected genes identified as Clock direct target genes that were associated with the indicated biological processes. H. Ridge plot of selected GO terms in flies expressing Clk^DN versus control at D1 identified using Gene Set Enrichment Analysis.

Clk^DN leads to progressive dysregulation of gene expression in photoreceptors at nearly a quarter of expressed genes.

To further identify direct versus indirect targets of the Clk:Cyc complex, we compared previously published high-confidence binding sites identified for Clk and Cyc using ChIP-seq in

*Drosophila* heads and bodies [51] with our list of accessible peaks obtained with ATAC-seq data from D10 photoreceptors. Genomic overlap analysis revealed that 10% of accessible peaks contained an experimentally identified Clk:Cyc TFBS (Fig 3D). Next, we annotated these photoreceptor-specific TFBS (n = 1461) to their nearest transcription start site (TSS) and found that 1004 photoreceptor-expressed genes contained a potential Clk:Cyc binding site. We refer to these 1004 genes as "Clock-bound genes" or CBGs (Fig 3E), which we compared to genes that were differentially expressed in Rh1>Clk$^{DN}$ relative to Rh1>Ctrl at either D1 or D10. We reasoned that if a gene is bound by Clk:Cyc and is differentially expressed, then it can be classified as a "direct target" of Clk:Cyc. Using these criteria, we identified 475 direct Clk:Cyc targets in photoreceptors. In contrast, we identified a far greater number of genes (2253) that are likely to be indirect targets of Clk:Cyc regulation, at least based on the available ChIP-seq data from whole heads [51]. Consistent with the predicted role of Clk:Cyc as transcriptional activators, Clk:Cyc direct targets were more likely to be downregulated relative to indirect targets upon expression of Clk$^{DN}$ (S3C Fig). GO term analysis of direct Clk:Cyc targets showed significant enrichment of genes associated with several biological processes, including phototransduction and circadian rhythm (Fig 3F). However, both direct and indirect Clk:Cyc targets were enriched for processes including TF activity and eye development. This is consistent with previous reports that have proposed that the Clk:Cyc complex acts at the top of a TF hierarchy, directly regulating transcript levels of key TFs in specific cell types, thereby indirectly regulating expression of their target genes [51,52]. Notably, individual inspection of the TFs that were classified as direct Clk:Cyc targets confirmed that several eye-specific TFs such as *ocelli* (*oc/otd*) and *eyeless* (ey) are bound at their promoters directly by Clk and Cyc (Fig 3G). To further identify the magnitude of changes in expression of these pathways upon disruption of Clk:Cyc, we performed gene set enrichment analysis comparing Clk$^{DN}$ vs Control at either D1 or D10 and obtained ridge plots showing the density of fold change for the genes associated with each pathway. Whereas upregulated genes showed significant enrichment of several biological processes, including TF activity, downregulated genes were associated with several phototransduction-associated pathways, including deactivation of rhodopsin signaling (Figs 3H, S3D and S3E).

Because light entrains the circadian clock to activate Clock-mediated transcription, we next performed RNA-seq in photoreceptors from Rh1>GFP$^{KASH}$ flies reared in constant darkness (DD). Under these conditions, we observed differential expression of genes associated with several biological processes, including phototransduction and circadian rhythm in dark raised flies relative to LD (S3F and S3G Fig). Because these processes were also enriched in the genes with differential expression upon disruption of Clk:Cyc activity, our data suggest that these biological processes are normally regulated by the circadian clock in photoreceptors in response to the light/dark cycle.

Taken together, these data show that the Clk:Cyc complex is a major transcriptional regulator of the photoreceptor transcriptome, including the phototransduction pathway. Additionally, our data suggest that Clk activity regulates gene regulatory networks by regulating expression of TFs that in turn direct expression of a large proportion of the transcriptome in photoreceptors.

## Expression of Clk$^{DN}$ leads to genome-wide changes in TF activity and a widespread decrease in chromatin accessibility

Because expression of Clk$^{DN}$ led to transcriptional dysregulation of TFs, we further investigated the TFs that showed changes in transcript level upon disruption of the Clk:Cyc complex. Gene Concept Network (cnet) plots demonstrated significant enrichment of 62 TFs amongst

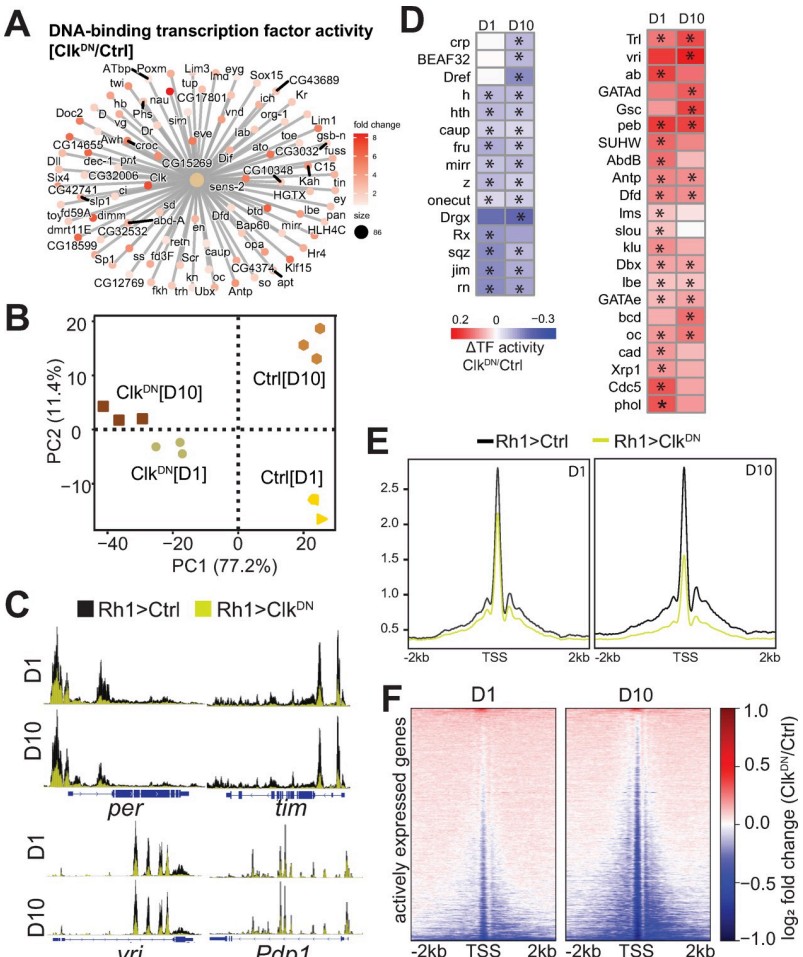

**Fig 4. Clock activity promotes changes in TF activity and maintains global levels of chromatin accessibility in photoreceptors.** A. Cnet plot of genes associated with the GO term "DNA-binding transcription factor activity". Color represents change in expression in Clk$^{DN}$ relative to control at D1. B. PCA of Clk$^{DN}$ and control ATAC-seq samples based on the read distribution over a 10-kb binned genome. C. Genome browser inspection of CPM-normalized ATAC-seq signal for selected core Clock genes (*per*, *tim*, *vri* and *Pdp1*). Scale is normalized to the same height in each comparison. ATAC-seq from Ctrl flies is labeled in black, and Clk$^{DN}$ in yellow. D. Hierarchical clustered heatmap of significant Clock-regulated transcription factors identified between Rh1>Clk$^{DN}$ and Rh1>Ctrl at either D1 or D10. Scale represents the relative change in TF activity. E. CPM-normalized gene metaplots of ATAC-seq signal centered around Transcription Start Sites (TSS). F. ATAC-seq fold change heatmaps of signal centered around TSS for all actively expressed genes in photoreceptors. Fold change is obtained by dividing signal from Rh1>Clk$^{DN}$ relative to Rh1>Ctrl, and log$^2$ transformed to center changes around zero (no change).

the upregulated genes in flies expressing Clk$^{DN}$ (Fig 4A). Based on these changes in TF expression, we wondered if expression of Clk$^{DN}$ would also lead to significant changes in TF activity relative to Rh1>Ctrl flies. To test this, we profiled chromatin accessibility in Rh1>Clk$^{DN}$ versus control, and integrated the RNA-seq and ATAC-seq data using the *diffTF* approach, as in Fig 2. PCA analysis of the accessible chromatin landscape revealed that 77% of the variation was explained by expression of Clk$^{DN}$ (Fig 4B). In addition, Clk:Cyc targets, such as *tim*, *per*, *vri* and *Pdp1*, exhibited significant decreases in accessibility through-out their gene bodies upon expression of Clk$^{DN}$ (Fig 4C). Importantly, genomic annotation of accessible peaks revealed a stable distribution of peaks in all samples (S4A Fig), suggesting that disruption of the Clk:Cyc complex does not lead to overall changes in the genome-wide distribution of

accessible peaks. Rather, these data show that the Clk:Cyc complex promotes chromatin accessibility at target genes, consistent with the well characterized role of Clk:Cyc in transcription activation [53].

Using *diffTF*, we identified 37 TFs with differential activity upon expression of Clk[DN], or "Clock-regulated TFs". Whereas 15 TFs had decreased activity, 22 showed increased activity in Rh1>Clk[DN] relative to Rh1>Ctrl (Fig 4D). Interestingly, several of the genes encoding these Clock-regulated TFs are also directly bound by Clk and Cyc at their promoters (see Fig 3), including the eye-specific TFs *oc/Otd* and *ey*, as well as *Xrp1*, *onecut*, *crp*, and *Trl*. We also identified increased TF activity of the circadian regulator *vri*, which represses transcription of *Clk* [54]. Consistent with the reported role of *vri* as a repressor, we observed decreased levels of wild-type *Clk* transcript in Rh1>Clk[DN] flies using qPCR with primers that differentiate between the wild-type and dominant negative version (see S2B Fig). Thus, these data suggest that the Clk:Cyc complex contributes to the maintenance of gene regulatory networks in photoreceptors by regulating transcript levels and/or TF motif/binding activity of many transcription factors.

Visual inspection of chromatin accessibility tracks showed that the Rh1>Clk[DN] flies had an overall decrease in chromatin accessibility around promoters and throughout gene bodies (Fig 4C). To further evaluate this, we obtained gene metaplots averaging the ATAC-seq signal around promoters for all actively expressed genes in Rh1>Clk[DN] and Rh1>Ctrl flies. Strikingly, we observed that expression of Clk[DN] led to a widespread decrease in chromatin accessibility around transcription start sites, and this decrease in global accessibility was exacerbated at D10 relative to D1 (Fig 4E). These gene-based observations were corroborated by peak-based quantification (S4B Fig). Thus, our data suggest that disruption of the Clk:Cyc complex results in a global decrease in chromatin accessibility in photoreceptors, even though Clk and Cyc only have 475 direct gene targets in this cell type (Fig 3E). To further gain insight if these changes in accessibility were progressive between D1 and D10 for a given gene, we obtained heatmaps of ATAC-seq fold change signal for actively expressed genes at either D1 or D10 in which both heatmaps were sorted identically (Fig 4F). These heatmaps revealed that genes that had decreased accessibility at D1 also showed sustained decreases in accessibility at D10, suggesting that disruption of the Clk:Cyc complex has a role in promoting chromatin accessibility in photoreceptors at a large fraction of actively expressed genes, although our data does not identify if this role is direct or indirect.

Taken together, our data shows that expression of Clk[DN] leads to dysregulation of TF levels and/or activity, suggesting the Clk:Cyc modulates gene regulatory networks associated with these TFs. In addition, expression of Clk[DN] leads to a widespread decrease in chromatin accessibility that is independent from changes in gene expression, suggesting that the circadian clock contributes to the global maintenance of chromatin remodeling of actively expressed genes in *Drosophila* photoreceptors.

### Disruption of Clock activity leads to light-dependent retinal degeneration and light-independent accumulation of oxidative stress

Disruption of the phototransduction machinery by mutations in phototransduction genes that leads to loss of function or decreased expression is often associated with retinal degeneration [55]. Consistent with the GSEA analysis (see Fig 3H), cnet plots evaluating RNA-seq fold change for genes associated with phototransduction revealed significant decreases in transcript levels of 22 genes in flies expressing Clk[DN] (Figs 5A and S5). These genes included many phototransduction components whose loss leads to light-dependent retinal degeneration, such as *Cds*, *Arr2*, *rdgB*, *trp*, *rdgC*, and *ninaC* (marked with a red asterisk, Fig 5A) [55]. Thus, we

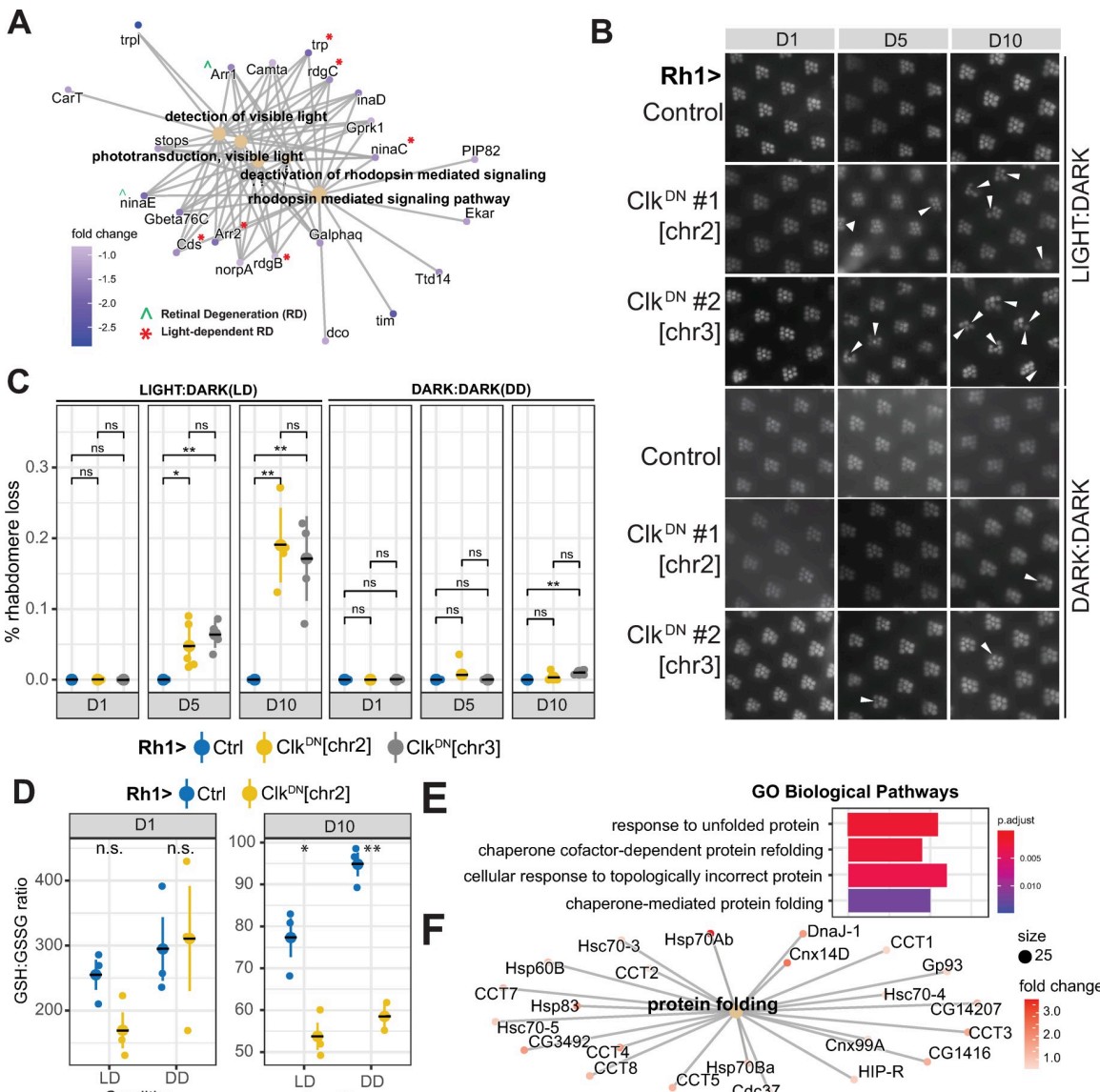

**Fig 5. Expression of Clk^DN leads to light-dependent retinal degeneration and light-independent accumulation of oxidative stress.**
A. Cnet plot of DEGs (p-adj<0.05,|FC|>1.5) in Rh1>Clk^DN relative to Rh1>Ctrl at D1, associated with phototransduction-related GO terms. Genes associated with light-dependent and -independent retinal degeneration based on published literature are indicated by accent or asterisks. B. Representative images of eyes from flies expressing Clk^DN versus control at the indicated age reared in light:dark (LD–top) or constant dark (DD—bottom) conditions. Images were obtained using optical neutralization (see Materials and Methods). Arrows indicate position of missing/degenerated rhabdomeres. C. Scatter plot showing quantification of retinal degeneration in panel B. Individual eyes are shown as circles with means shown as horizontal black lines overlaying large circle (n = 5). The distribution for each LD reared sample was compared with the DD control for the same age using ANOVA followed by Tukey-HSD comparing ages, samples, and condition (L:D vs D:D). "n.s." = not significant, "*" = FDR<0.05, "**" = FDR<0.005). D. Reduced(GSH):Oxidized(GSSG) Glutathione ratios in dissected eyes from flies of the indicated genotype, age, and condition. Statistics were performed as in panel C. E. Bar plots representing enriched GO terms amongst the genes that were upregulated in Rh1>Clk^DN at D10 relative to D1. F. Cnet plots of genes identified in panel E.

hypothesized that expression of Clk^DN in *Drosophila* photoreceptors would lead to retinal degeneration. To test this, we examined photoreceptor integrity using optical neutralization (see Materials and Methods) in Rh1>Clk^DN and Rh1>Ctrl flies that were maintained in either standard 12:12 hour light:dark (LD) or free-running, constant darkness (DD) conditions. We

note that all of the flies used in this study had pigmented eyes (S5B Fig), and were not susceptible to light-mediated damage as is the case for $w^{1118}$ flies. We found that Rh1>Clk$^{DN}$ flies presented progressive retinal degeneration starting at day 5, when raised under standard light: dark (LD) conditions relative to Rh1>Ctrl flies (Fig 5B top and 5C left). Importantly, photoreceptors were intact at earlier adult stages just after eclosion (D1), although qPCR analysis from 1-day old heads shows that the *Clk$^{DN}$* transcript is already expressed at this age (S3B Fig). We observed retinal degeneration in two independent *UAS-Clk$^{DN}$* lines inserted on different chromosomes, suggesting that retinal degeneration is unlikely to result from insertion position of the transgene; moreover, all experiments were performed in progeny hemizygous for the *UAS-Clk$^{DN}$* transgene. To test if the retinal degeneration resulting from Clk$^{DN}$ expression was dependent on light exposure, we raised Rh1>Ctrl and Rh1>Clk$^{DN}$ flies in constant darkness (DD) and monitored retinal degeneration. Strikingly, rearing Rh1>Clk$^{DN}$ flies in constant darkness prevented the onset of retinal degeneration in both Clk$^{DN}$ lines (Fig 5B-bottom and 5C-right) demonstrating that expression of Clk$^{DN}$ results in light-dependent retinal degeneration in adult *Drosophila* photoreceptors. Since Clk$^{DN}$ expression resulted in light-dependent retinal degeneration, we next asked if expression of the analogous dominant negative for its partner Cycle (Cyc$^{DN}$) caused a similar phenotype. Surprisingly, we did not observe any retinal degeneration in Rh1>Cyc$^{DN}$ flies reared in standard LD conditions at either D5 or D10 (S5C Fig). When we examined the relative level of *Clk* and *cyc* transcripts in photoreceptors, we found that *Clk* transcripts are 10-times more abundant than *cyc* (S5D Fig). In addition, proteomic analysis of fly heads during the day revealed that there are about 5-fold fewer peptides corresponding to Cyc versus Clk [48]. Thus, the differences in phenotype between Clk$^{DN}$ and Cyc$^{DN}$ flies could arise because of differences in protein abundance; however, it remains possible that Clk and/or Cyc have independent functions in photoreceptors outside of the canonical Clk:Cyc complex.

The circadian clock has also been shown to be necessary to protect cells against oxidative stress. For example, flies carrying the *Clk$^{Jrk}$* allele, which produces a mis-spliced *Clk* transcript and leads to decreased clock activity [56], show increased levels of reactive oxygen species (ROS) in aging brains [57]. Since light is a major source of oxidative stress in the eye [58] and Rh1>Clk$^{DN}$ showed light-dependent retinal degeneration, we hypothesized that Rh1>Clk$^{DN}$ flies exposed to light had increased oxidative stress levels relative to Rh1>Ctrl flies or flies reared in DD conditions. To test this, we performed a targeted metabolite assay to measure the ratio of reduced and oxidized glutathione (GSH:GSSG) in Rh1>Clk$^{DN}$ and Rh1>Ctrl flies reared in LD or DD conditions at D1 and D10. In this assay, a lower GSH:GSSG ratio is indicative of increased oxidative stress levels [59]. We note that given the technical limitations for isolation of intact photoreceptors, we performed these targeted metabolite assays using dissected eyes. Using this approach, we found that oxidative stress levels did not show significant changes at D1 in any condition (Fig 5D-left). Unexpectedly, we observed a significant decrease in the GSH:GSSG ratio in Rh1>Clk$^{DN}$ flies relative to Rh1>Ctrl flies raised in either LD and DD conditions at D10 (Fig 5D-right). Thus, our data shows that Clock activity protects the *Drosophila* eye against oxidative stress. However, since expression of Clk$^{DN}$ only caused retinal degeneration in flies reared in LD, it is unlikely that the increased ROS levels are responsible for the retinal degeneration observed in Clk$^{DN}$ flies at D10.

Together, these data suggest that the disruption in expression of the phototransduction machinery in photoreceptors that lack Clock activity is likely responsible for the light-dependent retinal degeneration that we observed upon expression of Clk$^{DN}$. When we performed GO term analysis of genes that were differentially expressed in Rh1>Clk$^{DN}$ at both D10 and D1, we observed significant upregulation of genes associated with response to unfolded protein and response to topologically incorrect protein, including many heat shock proteins and

chaperones (Fig 5E and 5F). When the light-sensing Rhodopsin 1 (Rh1) is not properly inactivated, photoreceptor neurons experience substantial protein misfolding and ER stress, which leads to retinal degeneration in a light-dependent manner [55]. Since many of the genes that regulate Rhodopsin folding and inactivation were downregulated in Rh1>Clk$^{DN}$ relative to Rh1>Ctrl flies, our data suggest that the circadian clock directly contributes to Rhodopsin maintenance in *Drosophila* photoreceptors. Importantly, the onset of light-dependent retinal degeneration associated with failures in Rh1 inactivation can be caused by several factors, including excessive endocytosis of Rh1 and increased Ca$^{2+}$ influx [55,60–62]. We propose that Clock activity is protective in the retina because it promotes expression of genes that contribute to proper recycling of Rh1 upon light-exposure. However, our data also uncover an additional neuroprotective role of Clock by contributing to the response to oxidative stress, which might be important for additional tissues that do not contain the phototransduction machinery. This secondary role of Clock in the cellular response to oxidative stress might become increasingly important in advanced age, since accumulation of ROS is one of the hallmarks of aging.

## Discussion

The circadian clock maintains daily biological rhythms by controlling the expression of target output genes, and is highly conserved between *Drosophila* and humans [41]. Since many genes are rhythmically expressed in the retina including most of the phototransduction machinery [51,63,64], these observations suggest that the circadian clock plays a role in homeostatic regulation of the *Drosophila* retina. Here, we report that retina-specific expression of a dominant negative mutant of Clk in *Drosophila* leads to progressive light-dependent retinal degeneration and oxidative stress, showing that the circadian clock is required to maintain *Drosophila* photoreceptor integrity. Importantly, this role for the circadian clock in maintaining visual health is conserved in mammals because retina-specific disruption of *BMAL1*, the mice homolog of *cyc*, accelerates the loss of cone viability and function in aging *chx10$^{Cre}$;Bmal1$^{Fl/Fl}$* mice, which otherwise show a normal lifespan [65]. Further, disruption of *BMAL1* leads to loss of spectral identity and integrity of cone cells in *Crx-Cre;Bmal1$^{Fl/Fl}$* [66], and the rat retina shows a circadian-dependent loss of resistance to light-mediated damage [67,68]. The circadian clock also plays a broader role in maintaining health during aging because mice deficient in *BMAL1* in all tissues have reduced lifespan and several symptoms of premature aging including cataracts and neurodegeneration [69–71]. Moreover, homozygous *Per$^{luc}$* mice exhibit age-dependent photoreceptor degeneration and premature aging of the retinal pigment epithelium [72]. Since the PER::LUCIFERASE fusion protein is wild type and is used as a model for studying circadian rhythms [73], this observation suggests that even very mild changes in the expression or function of core circadian clock regulators can negatively impact the health of the aging eye. Notably, disruption of circadian rhythms in the human eye contributes to glaucoma and is also implicated in development of myopia [74,75]. We note that while preparing this manuscript, a preprint showed that expression of Clk$^{DN}$ under the photoreceptor-specific trpl-Gal4 driver caused decreased phototactic behavior in young flies relative to an age-matched control [76], which is consistent with the rhabdomere loss observed in our study. However, rhabdomere integrity was not tested by the authors of this study. Therefore, our studies suggest that the *Drosophila* retina serves as a useful model to study circadian-dependent regulation of photoreceptor homeostasis. It is important to note that Clk and Cyc could also have Clock-independent roles, as shown previously for aging brains [57]. Therefore, it will be important for future studies to establish Clock-dependent and -independent functions of Clk and Cyc in fly photoreceptors.

How does disruption of the circadian clock in photoreceptors lead to light-dependent retinal degeneration? Our data suggest that although Clk-dependent transcription is necessary to prevent high levels of oxidative stress in the eye, this is not the proximal cause of retinal degeneration in young flies. Instead, we favor a model in which Clk:Cyc directly binds and activates expression of genes encoding the phototransduction machinery in photoreceptors, maintaining the continued expression of these proteins that have an integral role in photoreceptor structure and function. Numerous studies have demonstrated that complete loss of function of individual phototransduction genes results in retinal degeneration, often dependent on light exposure [55]. Our data show a significant and substantial decrease in transcript levels of multiple phototransduction genes, and we propose that the cumulative decrease in expression of their corresponding proteins causes the light-dependent retinal degeneration in flies expressing Clk[DN]. However, because disrupting Clk:Cyc activity in photoreceptors had widespread effects on gene expression and chromatin accessibility, we cannot exclude the possibility that other pathways such as autophagy [77], $Ca^{2+}$ signaling [78] and phosphoinositide metabolism [79] also contribute to the onset of light-dependent retinal degeneration.

The circadian clock has been implicated in the oxidative stress response in *Drosophila* [57,80] and in mammals [81]. Since the onset of many age-related eye diseases is particularly sensitive to disruptions in oxidative stress response [82], an increase in ROS levels in Rh1>Clk[DN] eyes suggests an overall neuroprotective role for the circadian clock regulators in the retina, one which might become increasingly important with advanced age. Under standard laboratory conditions, wild-type flies with pigmented eyes only begin to exhibit the first signs of retinal degeneration after D50 [83]. Thus, we hypothesize that the increase in Clk:Cyc activity in aging photoreceptors protects against retinal degeneration in part, by promoting expression of genes that combat oxidative stress. Supporting this hypothesis, several stress response genes exhibit cyclic expression patterns in the head only in older flies (D55), and this cyclic expression is dependent on both Clk and oxidative stress [63]. However, because these studies were performed in female white-eyed *w[1118]* flies [63], which already show substantial retinal degeneration by D15 to D30 [84], it remains to be elucidated if these age-associated changes in cyclic transcription also occur in flies with pigmented eyes.

Here, we identified an age-dependent increase in Clk and Cyc TF activity in photoreceptors via an unbiased integrative ATAC-seq and RNA-seq approach, which focused on identifying changes in TF activity during aging based on changes in chromatin accessibility around TF binding sites. Thus, our initial approach was not focused on circadian biology and the data was obtained from single time-point comparisons; samples were collected at ZT6 +/- 1 for aging experiments and at ZT9 +/- 1 for the Clk[DN] experiments. Because of these sampling differences, the increase in Clk:Cyc TF activity in aging could reflect a change in the phase of Clk:Cyc binding to DNA during the day, as shown for Monarch butterfly brains [85], an increase in amplitude of binding activity, or both. Supporting the latter possibility, circadian analysis of the transcriptome of aging fly heads showed both an increase in the amplitude and a slight shift in phase of *tim* and *per* expression, moving earlier in the day as flies aged [63]. Other studies have observed a decrease in protein levels of the Clock repressor PER with age in *Drosophila* photoreceptors but not in pacemaker neurons [86,87], suggesting that the increased Clk:Cyc activity observed in our study could in fact reflect an increase in DNA binding of Clk:Cyc in old flies mediated by decreased repression by the Per:Tim complex. Nonetheless, the mechanisms underlying the age-associated changes in the circadian clock and Clk:Cyc TF activity in fly photoreceptors remain to be elucidated.

Overall, our work identifies a central role for the circadian clock in regulating the photoreceptor transcriptome. We observed that Clk:Cyc contributes to the expression of many thousands of genes in adult photoreceptors, consistent with reports from mammalian cells showing

that Clock activity regulates the expression of 15% of expressed genes [53]. Clk and Cyc have been proposed to act at the top of a regulatory hierarchy to control widespread cyclic transcription in many cell types by regulating the expression of TFs [51,52]. Our data from photoreceptors are consistent with Clk:Cyc regulating expression of several important eye-specific TFs in *Drosophila* photoreceptors, suggesting a mechanism through which Clk and Cyc control expression of many genes indirectly. This is likely a widespread phenomenon across *Drosophila* tissues because gene regulatory network analysis of cyclic transcripts in brain, gut, Malphigian tubules, and fat bodies also identified many Clock-regulated TFs, including *h*, *hth*, *Mitf, and GATAd* [88].

Interestingly, *per* (*per$^{01}$*) and *tim* (*tim$^{01}$*) male mutants have extended lifespan, mediated by a loss of PER in intestine cells [89], suggesting that increased Clock activity in a particular tissue can lead to positive outcomes associated with health- and lifespan in a tissue- and sex-specific manner. Thus, identifying the molecular mechanism that underlie the age-associated changes in Clk:Cyc activity in aging photoreceptors and their impact on cellular homeostasis, which may be specific to the retina or other peripheral clocks in flies, will be an important area of research for future studies.

## Materials and methods

### Fly collection and maintenance

Rh1-Gal4>UAS-GFP$^{KASH}$ (*w$^{1118}$;; P{w$^{+mC}$ = [UAS-GFP-Msp300KASH}attP2, P{ry$^{+t7.2}$ = rh1-GAL4}3, ry$^{506}$*) [3] flies were maintained on standard fly food as previously described. For aging experiments, flies were collected in a 3 day window after eclosion and transferred to population cages. For Clk$^{DN}$ experiments, flies were collected in a 24 hour period and transferred to population cages. D1 corresponds to flies that were collected the first day immediately after eclosion. Flies for both aging and Clk$^{DN}$ experiments were maintained in a 25°C incubator with a 12:12 hour light:dark cycle, where relative Zeitgeber Time (ZT) 0 corresponds to when the light cycle begins. For aging experiments, male flies were collected and flash-frozen at ZT6 (-/+) 1 hour. For Clk$^{DN}$ experiments, male flies were collected and flash-frozen at ZT9 (-/+) 1 hour. We note that UAS-GFP$^{KASH}$ and additional tagged KASH or QUAS flies [22] are available at Bloomington Drosophila Stock Center (BDSC) for nuclei immunoprecipitation in different tissues (#92580 for the UAS-GFP$^{KASH}$ flies used in this study). Rh1-Gal4 (BDSC #8691), UAS-Clk$^{DN}$ [#1] (BDSC #36319), UAS-Clk$^{DN}$ [#2] (BDSC #36318) and UAS-LacZ (BDSC #8529) fly lines were obtained from BDSC. UAS-Cyc$^{DN}$ were generously provided by Dr. Daniel Cavanaugh (Loyola University)

### Chromatin accessibility and transcriptome analysis of photoreceptors

Nuclei immuno-enrichment (NIE), Omni-ATAC, quantitative PCR (qPCR), and RNA-seq were performed on male *Rh1-Gal4>UAS-GFP$^{KASH}$* flies at the indicated ages as previously described [22]. Briefly, for NIE experiments, we processed 400 age-matched male flies per replicate. Three independent biological replicates were obtained and analyzed for all RNA-seq and ATAC-seq samples. RNA-seq and ATAC-seq data analysis were performed as described in [22,83]; details specific to this study are described below. The following primers were used for qPCR: *Clk$^{DN}$* (5'- CGACAAGGATGATACAGATCAG-3' and 5'- ATTGCTGAGGAACG CAGGCT-3'); Clk$^{WT}$ (5'- GCGAGAAGAAGCGACGAGAT-3' and 5'- ATTGCTGAGGAAC GCAGGCT-3'); *eIF1alpha* (5'- GCTGGGCAACGGTCGTCTGGAGGC-3' and 5'- CGTCT TCAGGTTCCTGGCCTCGTCCGG-3'); *RpL32* (5'- GCTAAGCTGTCGCACAAATG-3' and 5'- CGTTGTGCACCAGGAACTT-3').

Note, we used the same reverse primer to detect either WT or DN Clk transcript by combining with corresponding Forward primer (Fwd).

RNA-seq fold change heatmap analysis: Heatmaps were obtained using $\log_2$ transformed fold change values obtained from pair-wise comparisons using DESeq2 [90]. For aging heatmaps, DEGs obtained from each comparison with D10 were pooled and de-duplicated. Plots were obtained using R (v4.0.3) package *pheatmap* (v1.0.12).

Differential TF activity using diffTF: DiffTF [27] analysis was performed using the default parameters: analytical approach (*nPermutations = 0*), integration mode using raw counts obtained from STAR *(—quantMode GeneCounts)* [91], narrow peaks obtained using MACS2 [92]. TFBS bed files were obtained from PWMScan [93] using the Aug 2014 BDGP Rel6 + ISO1 MT/dm6 target genome with the 353 available motifs from the CIS-BP library [29] with default parameters *(p-value<0.00001, Bg base composition 0.29,0.21,0.21,0.29)*. Identified TFs were classified as significant if they had an FDR lower than 0.001 in at least *one* of the pair-wise aging comparisons with D10, or in either D1 or D10 analysis for the Clk$^{DN}$ comparisons. Detailed protocols for NIE, RNA-seq, Omni-ATAC, ChIP-seq, and CUT&Tag are available at: dx.doi.org/10.17504/protocols.io.buiqnudw.

Clk ChIP-seq peak and photoreceptor ATAC-seq overlap: Previously published high-confidence peaks obtained from Clk and Cyc ChIP-seq data [51] were downloaded, and genomic coordinates were transformed from the *dm3* to *dm6* genome using the *LiftOver* tool from the UCSC Genome Browser website [94]. Genomic overlap was calculated using R (v4.0.3) packages *ChIPpeakAnno* (v3.24.2) and *GenomicRanges* (v.1.42.0).

Transcript per million (TPM) scatter plots: TPM values for each sample and replicate were obtained using TPMCalculator [95]. TPMs were averaged between biological replicates and used for scatter plot generation using R package *ggplot2* (v3.3.3). DEGs obtained with DESeq2 were colored on the TPM scatter plots.

## Optic neutralization

Optic neutralization and retinal degeneration quantification were performed as previously described [17,96]. Briefly, flies were glued to a glass slide using transparent nail-polish, and eyes imaged using bright-field microscopy. We note that optical neutralization using bright-field microscopy (with white light) is only possible with flies that have pigmented eyes, which is the case for all the flies tested in the present manuscript (See S5B Fig). Five biological replicates were analyzed for each sample, and retinal degeneration scores were assessed blindly.

## Targeted GSH:GSSG metabolomic assay

25 eyes per sample were collected from male flies of the indicated age, condition, and genotype (n = 3). Eyes were placed in a Covaris MicroTube with 110 μl of blocking solution (62.5 mg NEM, 3 mg EDTA, 5 mg $NaHO_3$ disolved in 2 mL of 3:2 parts Water/MeOH (v/v)). Once all eyes were added to the blocking solution 10 μl of 100ng/ul of each internal standard was added $^{13}C_2$-$^{15}N$-GSH-NEM and $^{13}C_4$-$^{15}N_2$-GSSG. Samples were homogenized using a Covaris Ultra Sonicator with the following settings: peak power: 200; duty factor: 10%; cycles per burst: 200; time: 300 seconds. Samples were then processed using an Agilent 1260 Rapid Resolution liquid chromatography (LC) system coupled to an Agilent 6470 series QQQ mass spectrometer (MS/MS) to analyze glutathione [97]. (Agilent Technologies, Santa Clara, CA). A Waters T3 2.1 mm x 100 mm, 1.7 μm column was used for LC separation (Water Corp, Milford, MA). The buffers were A) water + 0.1% formic acid and B) acetonitrile + 0.1% formic acid. The linear LC gradient was as follows: time 0 minutes, 0% B; time 2 minutes, 0% B; time 8 minutes, 30% B; time 9 minutes, 95% B; time 9.1 minutes, 0% B; time 15 minutes, 0% B. The flow rate was 0.3

mL/min. Multiple reaction monitoring was used for MS analysis. Data were acquired in positive electrospray ionization (ESI) mode. The jet stream ESI interface had a gas temperature of 350˚C, gas flow rate of 9 L/minute, nebulizer pressure of 35 psi, sheath gas temperature of 300˚C, sheath gas flow rate of 9 L/minute, capillary voltage of 4000 V in positive mode, and nozzle voltage of 1000 V. The ΔEMV voltage was 450 V. Agilent Masshunter Quantitative analysis software was used for data analysis (version 8.0).

## Supporting information

**S1 Fig. Pearson correlation aging RNA-seq.** Pearson correlation heatmap of gene expression profiles from nuclear RNA-seq of aging samples (n = 3). Scores between 0 and 1 shown in each box correspond to Pearson's r score.
(PDF)

**S2 Fig. ATAC-seq peak genomic distribution.** A. Bar plot showing the genomic distribution of accessible peaks identified in ATAC-seq data in Rh1>GFP$^{KASH}$ during aging. Promoter annotation was based on -/+ 2 kb around transcription start sites. B. DNA binding motif for Clk and Cyc derived from CIS-BP database.
(PDF)

**S3 Fig. DEG analysis in Clk$^{DN}$ photoreceptors.** A. Bar plot showing $\log_2$-transformed transcript per million (TPM) values for core Clock genes comparing Rh1>Clk$^{DN}$ and Rh1>Ctrl at either D1 (left) or D10 (right) (mean +- SD, n = 3, P-value obtained using t-test). B. Bar plot showing *Clk$^{WT}$* and *Clk$^{DN}$* transcript levels measured in heads from male flies of the indicated genotype. Transcript levels are normalized to the geometric mean of housekeeping genes *eIF-1α* and *RpL32* (mean ± SD; n = 3, P-value obtained using t-test). "ND" = not detected, "*" = p-value<0.05, "**" = p-value<0.005. C. Box plots showing the fold change values for genes identified as direct or indirect targets. Fold change is obtained using DESeq2. P-adjusted > 0.05, |FC|>1.5. P-value is obtained using Wilcoxon test. D,E. Complete ridge plot obtained from Gene Set Enrichment Analysis comparing differentially expressed genes in Rh1>Clk$^{DN}$ relative to Rh1>Ctrl at D1 (D) or D10 (E). F. Volcano plot showing differentially expressed genes from photoreceptors when flies are reared in free-running conditions (dark:dark) versus light: dark cycle. Significance is equivalent to $-\log_{10}$(adjusted p-value). Genes are colored based on changes in gene expression in D:D relative to L:D. Red is upregulated, and blue is downregulated. DEGs have p-adj < 0.05. G. Gene Ontology (GO) term analysis of genes that were significantly up-regulated (top) or down-regulated (bottom) in D:D relative to L:D.
(PDF)

**S4 Fig. ATAC-seq analysis in Clk$^{DN}$ photoreceptors.** A. Bar plot showing the genomic distribution of accessible peaks identified in ATAC-seq data from the indicated genotypes and ages. Promoter annotation was based on -/+ 2 kb around transcription start sites. B. Volcano plots representing the differentially accessible peaks in Rh1>Clk$^{DN}$ relative to Rh1>Ctrl. Differentially accessible peaks are defined as having a False Discovery Rate (FDR) < 0.05, and absolute fold change (|FC|) > 1.5.
(PDF)

**S5 Fig. Optic neutralization in Rh1>Cyc$^{DN}$ flies.** A. Heatmap showing RNA-seq fold change values for genes associated with phototransduction in flies expressing Rh1>Clk$^{DN}$ relative to Rh1>Ctrl at D1 or D10. Fold change values are represented inside each square, and color-labelled based on fold change score. B. Images corresponding to representative flies from the indicated genotypes used to assess retinal degeneration via optical neutralization. Only males

are shown, but females also showed similar eye pigmentation. C. Representative images of eyes from flies expressing Cyc$^{DN}$ at the indicated age reared in light:dark (LD) conditions. D. *Clk* and *cyc* transcript ratios in aging samples. Kruskal-wallis test was used to determine differences amongst all groups, and pair-wise comparisons were obtained using standard t-test. No comparison was significant.
(PDF)

**S1 Table. DEGs identified during aging.** Differentially expressed genes comparing each aging time point relative to D10 (n = 3).
(XLSX)

**S2 Table. DEGs identified in Clk$^{DN}$.** Differentially expressed genes comparing Rh1>Clk$^{DN}$ relative to Control (n = 3) at D1 or D10.
(XLSX)

**S3 Table. Gene list with Flybase IDs.** List of genes used in the manuscript, with corresponding abbreviations, and Flybase gene IDs
(XLSX)

**S4 Table. Raw data retinal degeneration quantification.**
(XLSX)

**S5 Table. Raw data redox glutathione measurements.**
(XLSX)

## Acknowledgments

We thank the Weake lab for their comments on the manuscript. We thank Dr. Judith Zaugg (EMBL) and Dr. Paul Hardin (Texas A&M) for reading and providing critical comments about the manuscript. We also thank Amber Jannasch for processing the metabolite samples. The authors acknowledge the use of the facilities of the Bindley Bioscience Center, a core facility of the NIH-funded Indiana Clinical and Translational Sciences Institute. We thank Dr. Daniel Cavanaugh for providing the Cyc$^{DN}$ flies.

## Author Contributions

**Conceptualization:** Juan Jauregui-Lozano.

**Data curation:** Juan Jauregui-Lozano.

**Formal analysis:** Juan Jauregui-Lozano.

**Funding acquisition:** Vikki M. Weake.

**Investigation:** Juan Jauregui-Lozano, Hana Hall, Sarah C. Stanhope, Kimaya Bakhle, Makayla M. Marlin.

**Methodology:** Juan Jauregui-Lozano.

**Project administration:** Vikki M. Weake.

**Supervision:** Vikki M. Weake.

**Visualization:** Juan Jauregui-Lozano.

**Writing – original draft:** Juan Jauregui-Lozano.

**Writing – review & editing:** Juan Jauregui-Lozano, Vikki M. Weake.

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
