## [Decision Letter · Decision Letter 0]

13 Oct 2021

Dear Dr Weake,

Thank you very much for submitting your Research Article entitled 'The Clock:cycle complex is a major transcriptional regulator of Drosophila photoreceptors that protects the eye from retinal degeneration and oxidative stress' to PLOS Genetics.

The manuscript was fully evaluated at the editorial level and by independent peer reviewers. The reviewers appreciated the attention to an important problem, but raised some substantial concerns about the current manuscript. Based on the reviews, we will not be able to accept this version of the manuscript, but we would be willing to review a much-revised version. We cannot, of course, promise publication at that time.

If you decide to revise the manuscript for further consideration at PLOS Genetics, please aim to resubmit within the next 60 days, unless it will take extra time to address the concerns of the reviewers, in which case we would appreciate an expected resubmission date by email to plosgenetics@plos.org.

[LINK]

We are sorry that we cannot be more positive about your manuscript at this stage. Please do not hesitate to contact us if you have any concerns or questions.

Yours sincerely,

John Ewer

Associate Editor

PLOS Genetics

John Greally

Section Editor: Epigenetics

PLOS Genetics

Reviewer's Responses to Questions

**Comments to the Authors:**

Reviewer #1: This is a nice manuscript reporting the a possible new role for clock genes/proteins in the regulation of photoreceptors functions. The experimental design is scientifically sound and the results support the conclusion reached by the authors.

I have a few minor comments I would like to see addressed by the authors.

Perhaps the authors should discuss a little bit more the photoreceptors in Drosophila and why they decide to focus on the outer photoreceptors

In reading the manuscript I was totally confused by the nomenclature used by the authors with respect to the gene and protein. sometime the genes are in Italian with the first letter in capital sometime there are not the some is true for protein. They authors need to be consistent and they must use the appropriate nomeclature for Drosophila.

The authors need to provide more details about the statistical analysis used to determine difference in the gene expression. The sample size of the sample (n=3) does not really provide assurance about the rigor and reproducibility of the data. Therefore it is important that the statistical methodology used to determine differences is well explained in the manuscript.

The authors should expand a little the section in which they are comparing the data to the mouse. For example a recent paper in the mouse has shown that removal of the Bmal1 gene lead to change in the spectral identity of the cone photoreceptors by modulating the expression of a gene coding for an hormone (Sawant OB, et al. 2017 The circadian clock gene Bmal1 controls thyroid hormone mediated spectral identity and cone photoreceptor function. Cell Rep 21:692–70).

Reviewer #2: In this study, Jauregui-Lozano and colleagues use a combination of photoreceptor-specific gene expression profiling, ATAC-seq, bioinformatic analysis and Drosophila genetics to describe transcriptomic changes associated with aging. They further focus on changes in transcription factor activities. They conclude that the circadian regulator Clock shows sustained increases in activity during aging. Furthermore, expressing a Clock dominant negative transgene affected the expression of many eye specific genes, and caused light-dependent retinal degeneration. These results support the idea that Clock is a major neuroprotective factor during photoreceptor aging.

Overall, this is a solid and thorough study. The authors have employed multiple genomic approaches and have effectively used bioinformatic tools to draw attention to the role of Clock in age-related transcriptome change and degeneration. The extensive datasets collected at regular intervals during aging will serve as a useful resource. The role of circadian clock regulators affecting neurodegeneration has been reported in other biological settings (e.g., PMID 22227001), but the authors’ unbiased approach places Clock as a major factor associated with age-related transcriptome change – which is significant. I do have some comments that could help the authors further improve the manuscript.

1. Why should Clock activity change as flies age? Is this because the circadian rhythm changes? Or is it because Clock is activated by damage/injury/stress independent of the circadian rhythm? Simple measurements of the circadian rhythm of the aging flies might provide useful clues.

2. In the discussion, the authors speculate that aging could change the phase of Clock activity, or the amplitude. I believe that this could be tested experimentally. For example, the authors could assess Clock target gene expression at multiple time points throughout the day. They could compare the phase and amplitude of Clock target genes in young versus old flies.

3. The authors rely on Clock DN expression to assess Clock’s role in neuroprotection. It would be nice to independently validate phenotypes of Clk DN through other means (just to rule out off-target or genetic background effects). For example, does Clock RNAi also cause light-dependent retinal degeneration?

4. Is there an effect of over expressing wild type Clock in photoreceptors? Would such conditions enhance neuroprotection? Alternatively, would it cause retinal degeneration because it disrupts the circadian clock?

5. The Methods section doesn’t describe in sufficient detail how the authors performed the retinal degeneration assay. Were the flies in the w+ or w- background? Were there similar amounts of eye pigments in control versus Clock DN flies? Was retinal degeneration assessed through immunostaining?

6. Figure 5B has representative images of eyes in control and Clock DN flies. The legend does not say whether this is an immunohistochemical image (or not), and what antibody was used.

Reviewer #3: In the paper entitled: „The Clock:cycle complex is a major transcriptional regulator of Drosophila photoreceptors that protects the eye from retinal degeneration and oxidative stress”, the Authors Juan Jauregui-Lozano et al. report analyses of gene expression changes in R1-R6 photoreceptors of Drosophila during aging. Although, there is lots of data presented in the paper, the obtained results are unclear for me. First of all the Authors analysed gene expression in flies from 10 days old (males and females?) until 60 days old, sampling them every 10 days. They have found that some genes are up-regulated and other are down-regulated and these are rather expected results. In this part of the study I did not learn about mechanisms of aging. In the second part of the paper the Authors analysed the role of the circadian clock on gene expression in R1-R6 in 1 day and 10 days old flies. In this part of the study it is unclear if the Authors tried to learn about the expression of clk and cyc levels or the effect of the circadian clock disruption in R1-R6 photoreceptors. If the last is true, studying the clock effects on gene expression only at one ZT time points is not enough. The Authors compare results from the first part of the study with the second part but they examined flies in different age and collected them at the different time point. Moreover, it is well known that many genes are clock-controlled and the disruption of the clock leads to changes not only at the level of transcription but also at posttranscriptional processes, translation, posttranslational processes and on many other processes in the cell. In case of photoreceptors, it is known that the phototransduction is controlled by the peripheral clock in photoreceptors and by light and degeneration of the retina results not only from disruption of the clock but also from DNA brakes induced by UV and white light.

There are also problems in terminology and it is unclear when the Authors write about the clock, genes clock and cycle and when about proteins CLK and CYC.

In my opinion the paper needs a major revision to make clear statements about aging of the photoreceptors R1-R6 and about the clock effects in aging. For the second part of the paper the Authors used only young flies, 1 and 10 days old, so the results do not show aging in the photoreceptors.

Minor comments:

1. Everywhere in the text it is not clear what does it mean “loss of Clock activity”, it is gene, protein or the circadian clock.

2. P. 4 The Authors “identified 61 TFs with substantial changes in activity during aging”, however, they have not analysed proteins but genes.

3. P. 4 According to the Authors: “Our data identify a novel neuroprotective role for the circadian clock in the Drosophila eye, and suggest that this role may become increasingly critical in advanced age to prevent retinal degeneration.” I disagree with this statement since degeneration of the retina may also depend on desynchronization of rhythms in different cells in the retina. Moreover, the Authors did not show that overexpression of clk and cyc in 10 days old flies protects the photoreceptors against degeneration.

4. The following paragraph title: “Clock activity promotes changes in TF activity and maintains global levels of accessibility in photoreceptors” is unclear. What does it mean ”activity” in both cases.

5. P. 13, para 2, l. 2 What does it mean “chromatin accessibility around gene bodies”?

6. P. 16 Discussion l. 3. The eye is not required for circadian behaviour in flies but the retina photoreceptors contribute to the circadian behaviour.

7. P. 16 Discussion l. 5. It is possible that the effect of the circadian clock disruption on retinal degeneration is not direct.

8. P.16 Discussion. In case of BMAL1 many functions have been found which are not clock-dependent.

9. Fig. 5 B Representative images of eyes…. Unfortunately, degenerative changes are not visible in this figure.

**Have all data underlying the figures and results presented in the manuscript been provided?**

Reviewer #1: Yes

Reviewer #2: Yes

Reviewer #3: Yes

PLOS authors have the option to publish the peer review history of their article (what does this mean?). If published, this will include your full peer review and any attached files.

Reviewer #1: No

Reviewer #2: No

Reviewer #3: No

---

## [Decision Letter · Decision Letter 1]

3 Jan 2022

Dear Dr Weake,

Thank you very much for submitting your Research Article entitled 'The Clock:Cycle complex is a major transcriptional regulator of Drosophila photoreceptors that protects the eye from retinal degeneration and oxidative stress' to PLOS Genetics.

The manuscript was fully evaluated at the editorial level and by three independent peer reviewers. You will see that two of the reviewers were satisfied with your revisions. However, the third reviewer raised substantial concerns about the current manuscript. Based on the reviews, we will not be able to accept this version of the manuscript, but we would be willing to review a much-revised version. We cannot, of course, promise publication at that time.

If you decide to revise the manuscript for further consideration at PLOS Genetics, please aim to resubmit within the next 60 days, unless it will take extra time to address the concerns of the reviewers, in which case we would appreciate an expected resubmission date by email to plosgenetics@plos.org.

[LINK]

We are sorry that we cannot be more positive about your manuscript at this stage. Please do not hesitate to contact us if you have any concerns or questions.

Yours sincerely,

John Ewer

Associate Editor

PLOS Genetics

John Greally

Section Editor: Epigenetics

PLOS Genetics

Reviewer's Responses to Questions

**Comments to the Authors:**

Reviewer #1: The authors have addressed all my concerns

Reviewer #2: The authors have expanded their Discussion and added Supplementary Information to adequately address all points raised during the first round of review. I have no other points to raise.

Reviewer #3: The manuscript has been improved, however, because of the experimental procedure it still does not show which processes in R1-R6 photoreceptors depend on Clk as a component of the molecular mechanism of the circadian clock and which ones on clock independent Clk function. There is also lack of experimental data explaining why expression of clk and cyc increases in old flies.

I also do not understand why in adult flies, genes involved in the eye development (sevenless - expressed in R7, eyeless) show an increase in the expression with aging.

In my opinion the manuscript still misses important information but contains too many speculations.

Minor Comments:

l. 273 Genes regulated by the Clk/Cyc heterodimer are called clock-controlled genes (ccg) but not clock-bound genes.

l. 415 There is several processes to protect cells against oxidative stress and the Authors should measure ROS level to examine oxidative stress but not one of those processes.

l. 425 The Authors write: “We propose that Clock activity is protective in the retina because it promotes expression 20 427 of genes that contribute to proper recycling of Rh1 upon light-exposure.” I my opinion there are no data in the manuscript supporting this statement.

l. 438 I disagree with the Authors’ statement that “the circadian clock plays a role in retinal homeostasis independent from maintenance of rhythmic behaviors”.

The rhythms in the retina are correlated with rhythms in behaviour. The difference between retinal and behavioural rhythms is that circadian rhythms in photoreceptors can be regulated by the pacemaker, peripheral clocks inside photoreceptors or by both.

**Have all data underlying the figures and results presented in the manuscript been provided?**

Reviewer #1: Yes

Reviewer #2: Yes

Reviewer #3: Yes

PLOS authors have the option to publish the peer review history of their article (what does this mean?). If published, this will include your full peer review and any attached files.

Reviewer #1: **Yes: **Gianluca Tosini

Reviewer #2: No

Reviewer #3: No

---

## [Decision Letter · Decision Letter 2]

6 Jan 2022

Dear Dr Weake,

Thank you very much for submitting your Research Article entitled 'The Clock:Cycle complex is a major transcriptional regulator of Drosophila photoreceptors that protects the eye from retinal degeneration and oxidative stress' to PLOS Genetics.

The revised manuscript was again evaluated by Reviewer 3, who has accepted most of your answers to their concerns. Yet, there still remain a couple of issues we must ask you to address, see the Reviewer 3 feedback below for details.

[LINK]

Yours sincerely,

John Ewer

Associate Editor

PLOS Genetics

John Greally

Section Editor: Epigenetics

PLOS Genetics

Reviewer's Responses to Questions

**Comments to the Authors:**

Reviewer #3: I accept the Authors' answers to my comments. The Authors's plan for the future studies sounds good, however, in my opinion their current results do not explain aging of the clock in the retina photoreceptors R1-R6 and how its aging affects aging of photoreceptors and their degeneration. The Authors report some changes and I can accept it as their first approach.

In addition the Authors should correct the first sentence in the Authors' Summary and instead "photoreceptor neurons" (l.43) write the retina outer photoreceptors.

**Have all data underlying the figures and results presented in the manuscript been provided?**

Reviewer #3: Yes

PLOS authors have the option to publish the peer review history of their article (what does this mean?). If published, this will include your full peer review and any attached files.

Reviewer #3: No

---

## [Decision Letter · Decision Letter 3]

8 Jan 2022

Dear Dr Weake,

We are pleased to inform you that your manuscript entitled "The Clock:Cycle complex is a major transcriptional regulator of Drosophila photoreceptors that protects the eye from retinal degeneration and oxidative stress" has been editorially accepted for publication in PLOS Genetics. Congratulations for this and for your perseverance!

Yours sincerely,

John Ewer

Associate Editor

PLOS Genetics

John Greally

Section Editor: Epigenetics

PLOS Genetics

Comments from the reviewers (if applicable):

Reviewer's Responses to Questions

**Comments to the Authors:**

Reviewer #3: I do not have more comments.

**Have all data underlying the figures and results presented in the manuscript been provided?**

Reviewer #3: Yes

PLOS authors have the option to publish the peer review history of their article (what does this mean?). If published, this will include your full peer review and any attached files.

Reviewer #3: No

**Data Deposition**

http://datadryad.org/submit?journalID=pgenetics&manu=PGENETICS-D-21-01242R3

**Press Queries**

---

## [Editor Report · Acceptance letter]

26 Jan 2022

PGENETICS-D-21-01242R3 

The Clock:Cycle complex is a major transcriptional regulator of Drosophila photoreceptors that protects the eye from retinal degeneration and oxidative stress 

Dear Dr Weake, 

We are pleased to inform you that your manuscript entitled "The Clock:Cycle complex is a major transcriptional regulator of Drosophila photoreceptors that protects the eye from retinal degeneration and oxidative stress" has been formally accepted for publication in PLOS Genetics! Your manuscript is now with our production department and you will be notified of the publication date in due course.

With kind regards,

Zsanett Szabo

PLOS Genetics

On behalf of:
